# Development of a Low-Cost Fault Detector for Photovoltaic Module Array

**Kuei-Hsiang Chao \*, Jen-Hsiang Tsai and Ying-Hao Chen**

Department of Electrical Engineering, National Chin-Yi University, No.57, Sec. 2, Zhongshan Rd., Taiping Dist., Taichung 41170, Taiwan; anshar97@gmail.com (J.-H.T.); fishs0224@hotmail.com (Y.-H.C.)
**\*** Correspondence: chaokh@ncut.edu.tw; Tel.: +886-4-2392-4505 (ext. 7272)

**Abstract:** This study proposes a binary search-based fault detection system for photovoltaic (PV) modules to ameliorate the deficiencies in the existing fault detectors for PV module arrays. The proposed system applies a single-chip microcontroller to execute the binary search algorithm. Moreover, to overcome multi-node voltage detection and reduce the number of integrated circuit components, an analog switch is used to perform detection channel switching; the detection results are displayed on a software platform developed using Visual C#. The proposed system does not require learning to execute the fault diagnosis of PV module arrays and has advantages including high accuracy and low construction costs. Finally, to verify the feasibility of the proposed system, this study simulated the abnormal situations of actual modules and applied the binary search algorithm for maximum power point tracking to detect malfunctions of the PV module arrays.

**Keywords:** Fault diagnosis in photovoltaic (PV) module; maximum power point tracking (MPPT); photovoltaic (PV) module simulator; binary search algorithm; Visual C#

## 1. Introduction

To enhance power generation efficiency, photovoltaic (PV) systems are generally installed at unshaded locations. However, if such systems are installed outdoors over a long period, the PV modules might be degraded with time, engendering power reduction or failure; the output power of the systems substantially decreases in the event of shading or failure. Because detecting PV module failure through unassisted vision is difficult, conventionally examining each module through manually operated instruments would be a time-consuming process [1]. Therefore, establishing an effective fault diagnosis system is necessary.

Currently, numerous researchers are studying fault detection in PV systems. A previous study proposed a binary system coding method for detecting abnormal overcurrent situations in module arrays, thereby determining whether the modules are shaded [2]; subsequently, an overlapped diode can be connected to the modules to prevent them from the effects of shading. This method focuses only on detecting the effect of overcurrent on modules. However, overcurrent is not the sole factor associated with shading and faults in PV module arrays. The data measurement and calculation processes in this method are complicated; therefore, operators must possess a considerable level of expertise in order to execute this method, thus imposing a challenge in implementation. Another researcher utilized a thermal imager for fault diagnosis [3]. The researcher installed an additional thermal camera for detecting the temperature distribution of module arrays, thereby inferring the location of faults. However, when this approach is used in regions with relatively low irradiance, the temperature variation might not be substantial, resulting in reduced judgment accuracy. Moreover, the number of thermal cameras increases with the implementation area, raising the cost of large-scale PV power plant construction. Studies conducted on fault diagnosis have included additional sensors [4,5]

for extracting characteristic parameters for fault detection. Nevertheless, such sensors may deteriorate due to prolonged exposure to outdoor environments, and the adoption of sustainable sensors increases development costs. To avoid the deterioration of sensors or the increase in construction costs, the author of this paper discarded the idea of installing sensors outdoors and previously developed a detection circuit with an artificial intelligence algorithm for fault detection [6]; the disadvantage of this measure is that it can only determine which array in the module arrays is faulty, rather than the precise piece of faulty module.

A previous study proposed a MATLAB-based modeling and simulation scheme for analyzing PV module arrays under shaded conditions [7]. In this scheme, the variations of current–voltage (I–V) and power–voltage (P–V) characteristic curves are analyzed according to the number of shaded modules, configuration of PV module arrays, and whether a bypass diode or a blocking diode is installed. Finally, the relevant simulation parameters constitute the basis for determining shading or failure. In practice, the parameters of a PV module change when a failure occurs, resulting in imprecise fault output characteristics that yield inaccurate results.

Thus, to ameliorate the aforementioned deficiencies, this paper proposes a PV module fault diagnosis system that is based on a binary search algorithm [8]. This system accurately locates faulty modules and also requires only a low number of voltage sensors; thus, it has a low construction cost.

## 2. Maximum Power Point Tracking Techniques for PV Modules

### 2.1. Existing Maximum Power Point Tracking Techniques

To maintain the output power of a PV system at the maximum power point (MPP), maximum power point tracking (MPPT) techniques are commonly employed, such as variable universe fuzzy logic control considering temperature variability [9], power feedback method [10,11], perturb and observe method [12], and incremental conductance method [13]. This study integrated particle swarm optimization (PSO) with an existing conventional MPPT technique to prevent the output characteristic curve from producing more than two peaks, which might produce results that are constrained in local maxima and fail to track the actual MPP [14–16], when shading or failure occurs in PV module arrays. The proposed PSO algorithm employs a digital signal processor combined with a boost converter to achieve MPPT. Finally, failure cases were examined to verify the feasibility of the proposed method ensuring that a PV module array can operate at the MPP even with faulty modules at various locations.

### 2.2. PSO-Based MPPT Technique

PSO, an artificial intelligence algorithm, was proposed by Kennedy and Eberhart in 1995 [17]. This algorithm is inspired by information transmission associated with the foraging behaviors of birds. The algorithm comprises two essential parameters, namely the particle best value ($P_{best}$) and global best value ($G_{best}$), which influence the particle's direction and its search direction, respectively. Each bird is regarded as a particle, and any particle possesses the memory and experience generated during its own movement. A particle can learn to adjust its own movement according to its individual experience and memory. In PSO, several particles move simultaneously, and a specific particle compares its experience with the experience of other particles simultaneously to determine the local optimal solution. Because of the characteristics of PSO, particles can learn and memorize the evolution of swarms in addition to their own evolution, thus assisting them in searching for the best value. The schematic of the PSO search approach is shown in Figure 1.

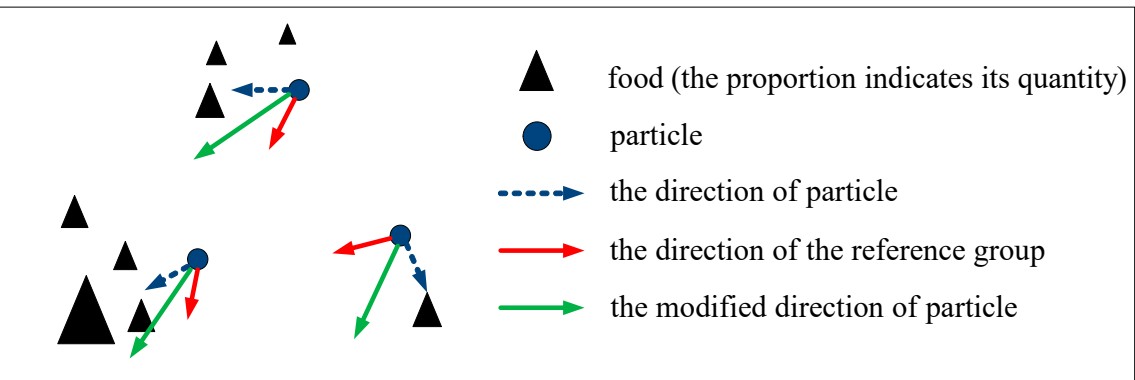

**Figure 1.** Schematic of PSO search approach.

The operating procedures of the PSO-based MPPT algorithm are outlined as follows:

Step 1: The number of particles ($i_{total}$) is set to 4, iteration number ($j_{total}$) is set to 100, weight value ($W$) is set to 0.4, individual learning factor ($C_1$) is set to 2, and group learning factor ($C_2$) is set to 2.

Step 2: The duty cycle is set as the position of a particle, and the desired objective function is established (i.e., the output power value of the PV module array).

Step 3: The present voltage, current, and power value of the PV module array are obtained after the position of the particle is set in step 2.

Step 4: The fitness function values of the particles are compared to determine the best position in order to update the best position a particle has memorized ($P_{best}$).

Step 5: $P_{best}$ is compared with $G_{best}$. If $P_{best}$ is superior to $G_{best}$, the current $G_{best}$ value is replaced by the $P_{best}$ value.

Step 6: Equations (1) and (2) of the PSO algorithm are used to update the parameter of each particle:

$$V_i^{j+1} = W \times V_i^j + C_1 \times rand1(\bullet) \times (P_{best,i} - P_i^j) + C_2 \times rand2(\bullet) \times (G_{best} - P_i^j) \tag{1}$$

$$P_i^{j+1} = V_i^{j+1} + P_i^j \tag{2}$$

In the preceding two equations, $V_i^j$ and $V_i^{j+1}$ denote the speeds of particle $i$ at time points $j$ and $j+1$, respectively; $W$ denotes the inertia weight; $C_1$ denotes the cognitive learning factor; $C_2$ denotes the social learning factor; $rand1 (\bullet)$ and $rand2 (\bullet)$ denote randomly generated values in the range of 0–1; $P_{best,i}^j$ denotes the best value for individual $i$ at $j$; and $G_{best}$ denotes the best fitness value of the swarm. In addition, $P_i^{j+1}$ and $P_i^j$ represent the positions of $i$ at $j+1$ and $j$, respectively.

Step 7: Steps 2–6 are repeated until the maximum number of iterations is attained.

Step 8: An assessment is conducted to determine whether the shading or failure condition changes. If the condition changes, the previous steps are repeated starting from step 1; otherwise, the ultimate value is retained.

## 3. Fault Detection Based on Binary Search Algorithm

In computer science, a binary search refers to a search algorithm that searches for a particular element in an ordered array. The search process starts from the middle elements of the array. If a middle element matches a target element, the search process is terminated. If the value of the target element is higher or lower than that of the middle element, the search continues in the half of the array that contains the element with the higher or lower value, respectively. Subsequently, the search continues by comparing the middle element of the array with the target element. In each comparison, the search method reduces the search range by half. Therefore, regardless of the number of series-parallel PV

modules, the operating procedures of the proposed binary search method are the same. The operating procedures of the binary search method are outlined as follows:

Step 1: Figure 2 illustrates a $4 \times 3$ series-parallel module array used for testing. Each of the 12 modules in the PV module array is assigned a code from 1 to 12 for identifying abnormal modules during detection. Subsequently, the arrays are divided into six regions, namely $L1$, $L2$, $L3$, $R1$, $R2$, and $R3$ (as shown in Figure 3), and each voltage node between the modules is assigned a code from $A$ to $K$.

Step 2: The voltage relationships among the regions $L1$ and $R1$, $L2$ and $R2$, and $L3$ and $R3$ are compared to identify regions in which abnormal modules may occur. The voltage of each region can be determined from the voltage between nodes (i.e., $B$, $E$, $H$, $J$ and $K$); the calculation procedure is presented in Equation (3), where $V_{ref}$ is the reference voltage of each region without module failure. If the voltage of a certain region is equal to $V_{ref}$, the region does not contain malfunctioning modules.

$$
\begin{aligned}
L1 &= V_{BK}, R1 = V_{JB}, \\
L2 &= V_{EK}, R2 = V_{JE}, \\
L3 &= V_{HK}, R3 = V_{JH}, \\
V_{ref} &= \frac{V_{JK}}{2}
\end{aligned}
\tag{3}
$$

Step 3: Each region's voltage data calculated using Equation (3) are then used to determine the search direction according to Table 1. If the direction cannot be identified, the module array is not malfunctioning or under full shading. However, the probability of full shading (or full failure) is approximately 0; therefore, it is not considered.

Step 4: After the search direction is determined through Table 1, the voltage of each module within the region involving a failure is calculated. For example, if the fault is type 1, then the voltages of module 1 and module 2 in $L1$—which might contain a faulty module—are calculated ($V_{module1} = V_{AK}$ and $V_{module2} = V_{BA}$, respectively). For the remaining fault types, the voltage of each module within the region involving a fault is calculated according to the same principle.

$$
\begin{aligned}
L1 &: V_{module1} = V_{AK}, V_{module2} = V_{BA} \\
R1 &: V_{module3} = V_{CB}, V_{module4} = V_{JC} \\
L2 &: V_{module5} = V_{DK}, V_{module6} = V_{ED} \\
R2 &: V_{module7} = V_{EF}, V_{module8} = V_{JF} \\
L3 &: V_{module9} = V_{GK}, V_{module10} = V_{HG} \\
R3 &: V_{module11} = V_{IG}, V_{module12} = V_{JI}
\end{aligned}
\tag{4}
$$

Step 5: The voltages of modules within the region involving potential faults are compared; the module with a relatively low voltage is considered abnormal. If the search direction covers more than two regions, the voltages of the modules in each region are directly compared, and the module with a lower voltage is considered faulty. The flow chart of the binary search algorithm is shown in Figure 4.

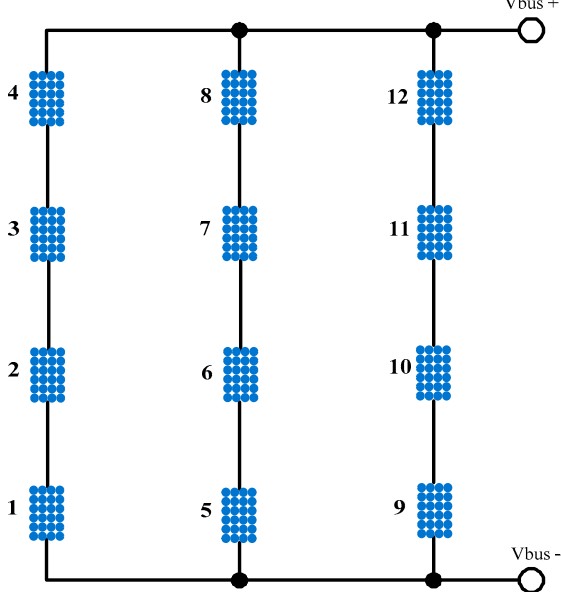

**Figure 2.** Illustration of 4 × 3 series-parallel PV module array.

**Table 1.** Search direction conditions.

| Failure Type | Voltage of Each Region | Search Direction | Failure Type | Voltage of Each Region | Search Direction |
|---|---|---|---|---|---|
| 1 | $L1 < R1$; $L2 \fallingdotseq R2 \fallingdotseq V_{ref}$; $L3 \fallingdotseq R3 \fallingdotseq V_{ref}$ | Search in the direction of $L1$ | 8 | $L1 > R1$; $L2 \fallingdotseq R2 \fallingdotseq V_{ref}$; $L3 > R3$ | Search in the direction of $R1$ and $R3$ |
| 2 | $L1 > R1$; $L2 \fallingdotseq R2 \fallingdotseq V_{ref}$; $L3 \fallingdotseq R3 \fallingdotseq V_{ref}$ | Search in the direction of $R1$ | 9 | $L1 \fallingdotseq R1 \fallingdotseq V_{ref}$; $L2 \fallingdotseq R2 \fallingdotseq V_{ref}$; $L3 < R3$ | Search in the direction of $L3$ |
| 3 | $L1 \fallingdotseq R1 \fallingdotseq V_{ref}$; $L2 < R2$; $L3 < R3$ | Search in the direction of $L2$ and $L3$ | 10 | $L1 \fallingdotseq R1 \fallingdotseq V_{ref}$; $L2 \fallingdotseq R2 \fallingdotseq V_{ref}$; $L3 > R3$ | Search in the direction of $R3$ |
| 4 | $L1 \fallingdotseq R1 \fallingdotseq V_{ref}$; $L2 > R2$; $L3 > R3$ | Search in the direction of $R2$ and $R3$ | 11 | $L1 < R1$; $L2 < R2$; $L3 \fallingdotseq R3 \fallingdotseq V_{ref}$ | Search in the direction of $L1$ and $L2$ |
| 5 | $L1 \fallingdotseq R1 \fallingdotseq V_{ref}$; $L2 > R2$; $L3 \fallingdotseq R3 \fallingdotseq V_{ref}$ | Search in the direction of $R2$ | 12 | $L1 > R1$; $L2 > R2$; $L3 \fallingdotseq R3 \fallingdotseq V_{ref}$ | Search in the direction of $R1$ and $R2$ |
| 6 | $L1 \fallingdotseq R1 \fallingdotseq V_{ref}$; $L2 < R2$; $L3 \fallingdotseq R3 \fallingdotseq V_{ref}$ | Search in the direction of $L2$ | 13 | $L1 < R1$; $L2 < R2$; $L3 < R3$ | Search in the direction of $L1$, $L2$, and $L3$ |
| 7 | $L1 < R1$; $L2 \fallingdotseq R2 \fallingdotseq V_{ref}$; $L3 < R3$ | Search in the direction of $L1$ and $L3$ | 14 | $L1 > R1$; $L2 > R2$; $L3 > R3$ | Search in the direction of $R1$, $R2$, and $R3$ |

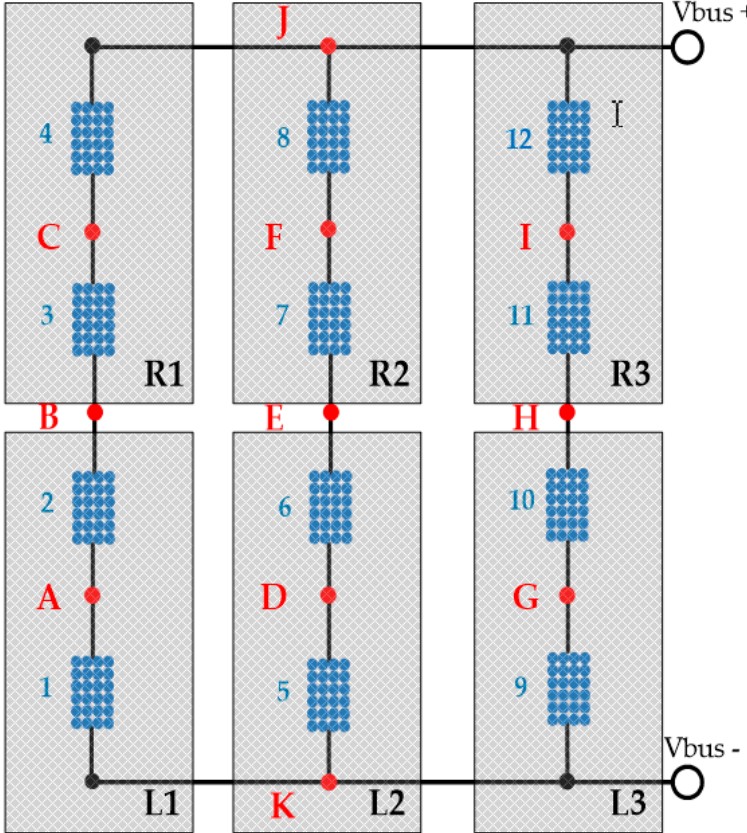

**Figure 3.** Regional division of 4 × 3 series-parallel PV module array.

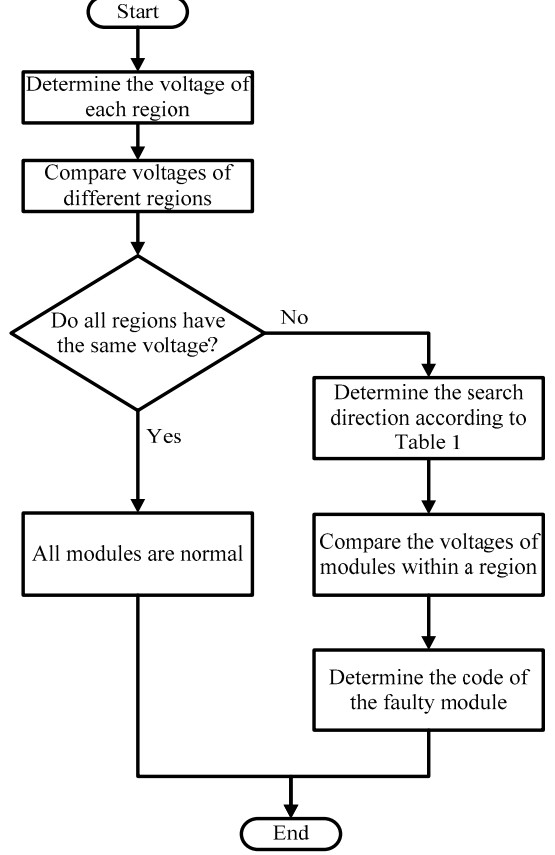

**Figure 4.** Flow chart of binary search algorithm.

## 4. Establishment of Fault Diagnosis Framework for PV Modules

Because weather conditions change constantly, it is difficult to reproduce the testing under the exact same weather conditions. Thus, it is a disadvantage for conducting fault diagnosis repeatedly under certain weather conditions anytime. Therefore, researchers must establish an MPPT system, fault detector, a software platform developed using Visual C#, and a PV module simulator applicable to examining various shading and fault conditions. In addition, Solar Pro simulation software can be applied to verify the corresponding electrical characteristics.

### 4.1. Development of MPPT System

MPPT is conducted using PSO because it can overcome the problem of multi-peak P–V curves. This problem occurs under failure or shading conditions and can prevent the detection of the actual MPP [18]. Figure 5 shows the framework of the PSO-based MPPT system.

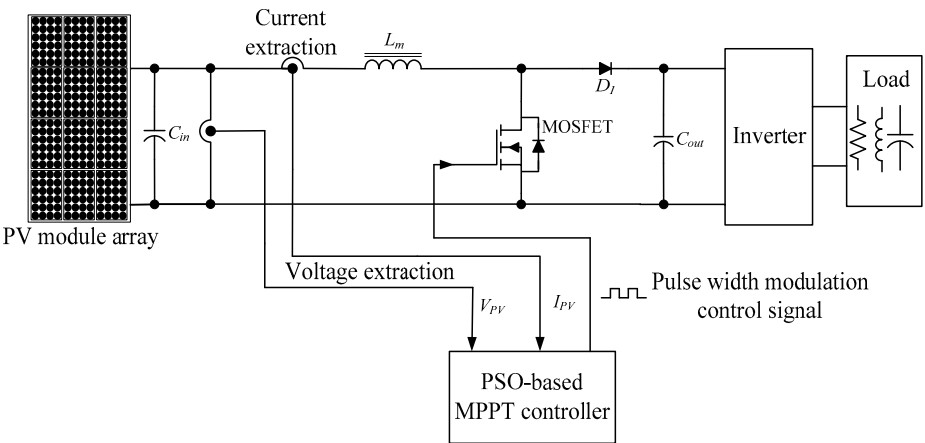

**Figure 5.** Framework of PSO-based MPPT system.

### 4.2. Development of Fault Detector

This study adopted a low-cost 8-bit microcontroller PIC18F8720 manufactured by Microchip Technology Inc. [19] to implement an online fault detection method. The detector applies a binary search algorithm to solve the problem of multi-node voltages. Moreover, analog switches are used to switch the detection channels to reduce the number of integrated circuit components. The test results are displayed on a software platform developed using Visual C#. Figure 6 presents the overall framework of the system.

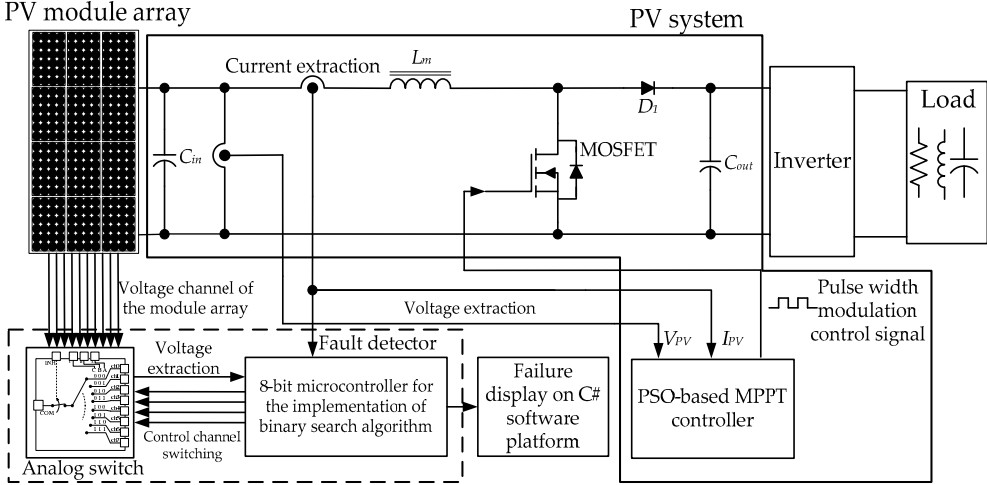

**Figure 6.** Framework of binary search-based fault diagnosis system for PV modules.

### 4.3. Software Platform Developed Using Visual C#

Figure 7 depicts the interface of fault diagnosis software developed using Visual C#. Users should select the appropriate communication port before initiating a monitoring process; after this selection, the detection result is displayed in the detection result section. If a module failure occurs in the PV module array, the color of the faulty module turns from blue to pink to notify the user (Figure 8). In addition, the system automatically sends the detection result to the user through e-mail if the automatic alert box is checked and the user provides an e-mail address in advance (Figure 9).

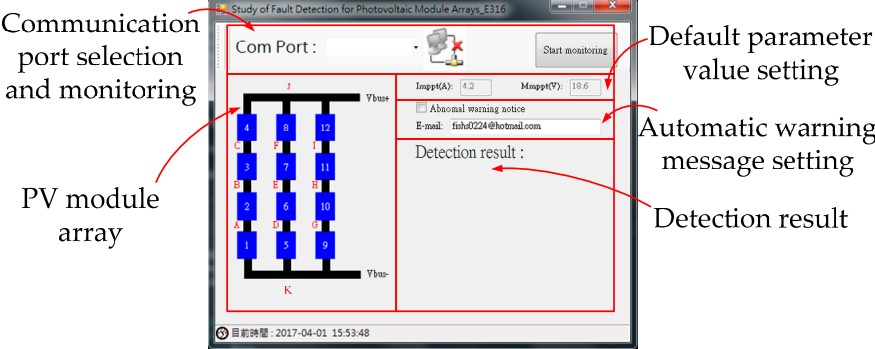

**Figure 7.** Interface of fault diagnosis software.

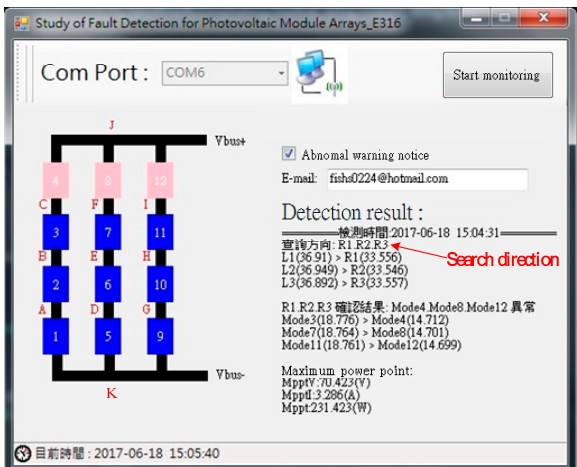

**Figure 8.** Actual detection status of the fault diagnosis software interface.

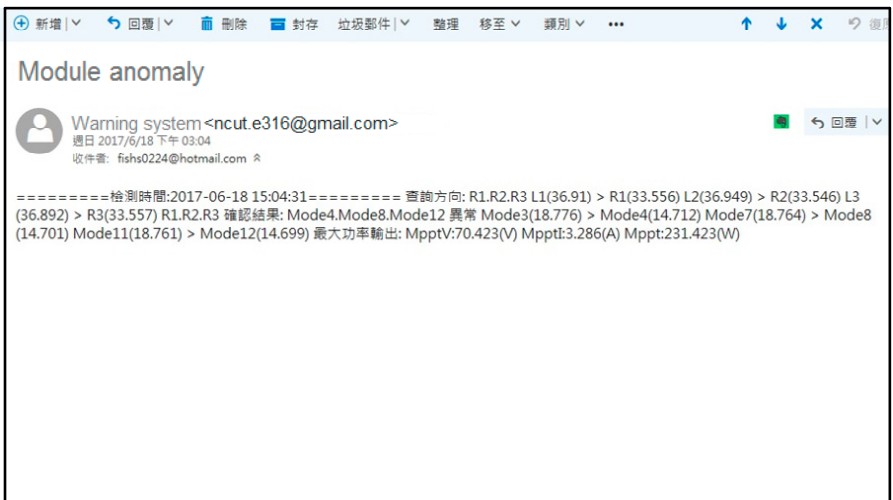

**Figure 9.** Automatic warning message.

### 4.4. Establishment of PV Module Simulator with Adjustable Settings for Shading and Failure

The SANYO HIP-2717 PV module simulator can be set for shading and fault conditions. Figure 10 illustrates the circuit framework of the simulator. The value of the open-circuit voltage and short-circuit current of the simulator can be adjusted to reflect the electrical characteristics of various shade conditions. In the simulator, shading was set to 0% and 30%. Figures 11 and 12 present comparisons of I–V and P–V curves measured by the EKO MP-170 I–V checker and those plotted using the Solar Pro simulation module at an irradiance of 1000 W/m$^2$ (0% shading) and 700 W/m$^2$ (30% shading), respectively. The characteristic curve of the proposed simulator is shown to conform to the characteristics of the SANYO HIP-2717 module. Based on the analysis in Reference [20], it can be seen that the open-circuit voltageof the module at normal operation and shaded condition are calculated in a different way. Therefore, the characteristics of I-V and P-V curves will be different under normal operation and shading conditions. This will result in underestimation at 0% shading and overestimation at 30% shading. However, the curve trends of the simulator are consistent with that of a real photovoltaic module.

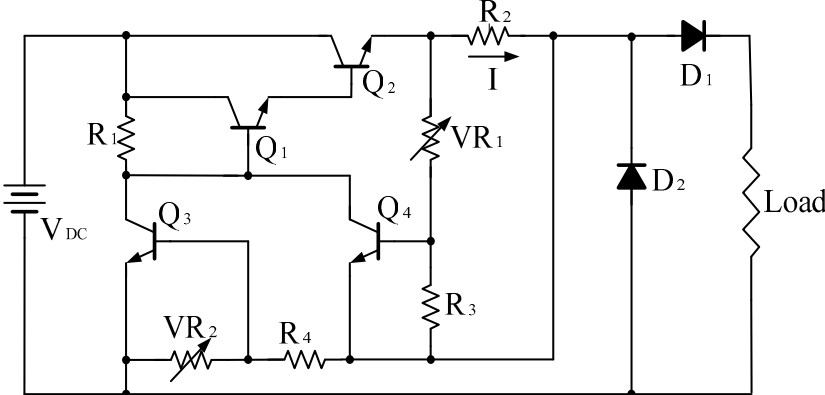

**Figure 10.** Circuit of the PV module simulator with adjustable settings for shading and failure.

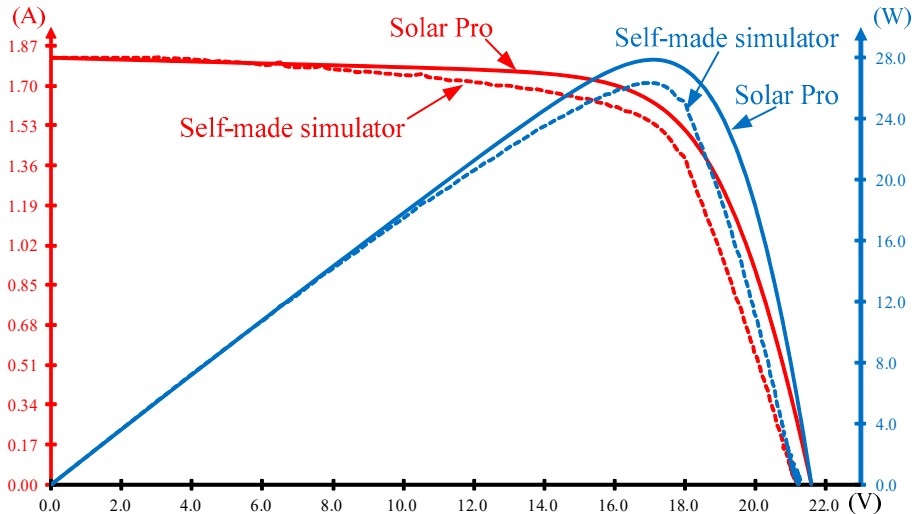

**Figure 11.** I-V and P-V characteristic curves of Solar Pro simulation and self-made simulator under 0% shading.

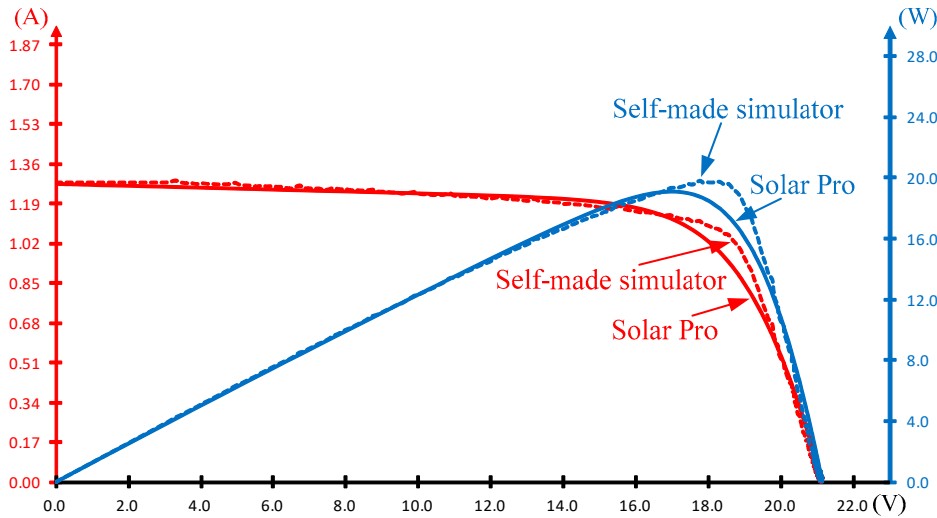

**Figure 12.** I-V and P-V characteristic curves of Solar Pro simulation and self-made simulator under 30% shading.

### 4.5. Establishment of Technology for Processing Information Obtained from Testing

The adopted microcontroller uses a universal asynchronous receiver–transmitter (UART) as the channel for information output, transmitting the measured data to the personal computer (PC) terminal for subsequent processing. However, the PC only has a USB serial port and cannot directly link with the UART interface of the microcontroller. Therefore, a signal conversion tool, namely PL2303, is required to serve as the conversion interface between UART and USB. Figure 13 shows the signal conversion relationship. The direct current voltage ranges of the PC and microcontroller terminals are dissimilar. Therefore, because of safety considerations, an isolation system is necessary during signal conversion between the two. Figure 14 illustrates a self-designed isolation circuit that contains a high-speed optical coupler (6N137) for signal isolation during conversion.

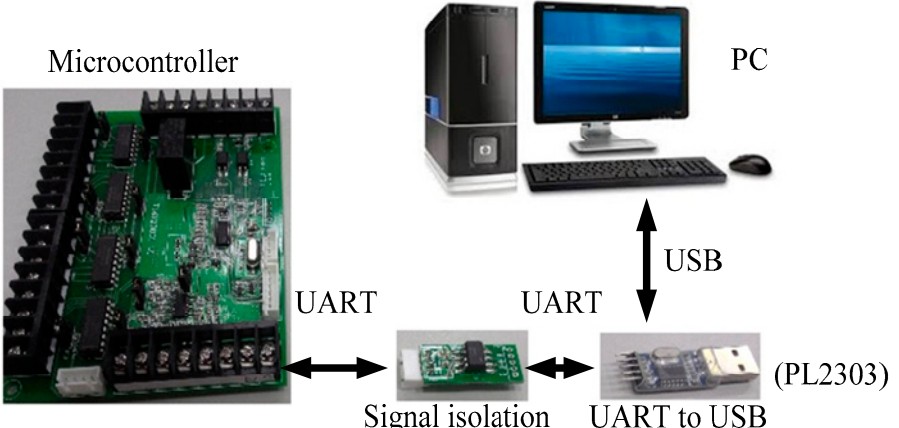

**Figure 13.** Signal conversion diagram.

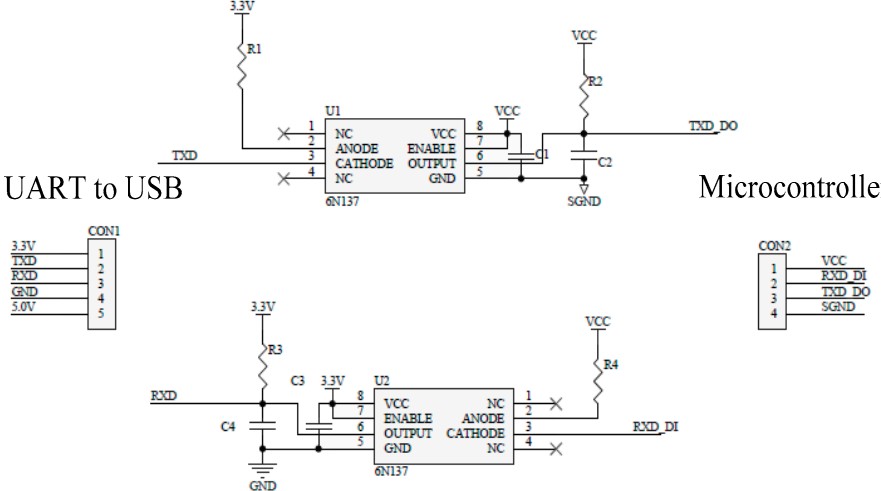

**Figure 14.** Circuit of signal isolation.

## 5. Measurement Results

The modules of the self-made PV simulator were arranged in a 4 × 3 series-parallel configuration to form a PV module array that can adjust shading proportions and failure conditions (Figure 15). The module array applied the PSO algorithm to conduct MPPT; thus, the PV module array operated at MPP. Subsequently, the voltage values of the related nodes in the array were extracted for fault diagnosis using the proposed fault detector. Through the communication interface, the test results were exported to the software platform developed, using Visual C# for display. The proposed software can not only accurately detect the faulty module but also provide a schematic of the module array, with the color of the faulty module being converted from blue (the original color) to pink. Such a display system enables a direct interpretation of the results. Moreover, the failure types were further divided into 14 possible states as shown in Table 1. The experimental outcomes of these nine failure types are shown in Figures 16–24, respectively. The results indicate that the proposed fault detector can accurately locate faulty modules and let the PV module array operate at MPP.

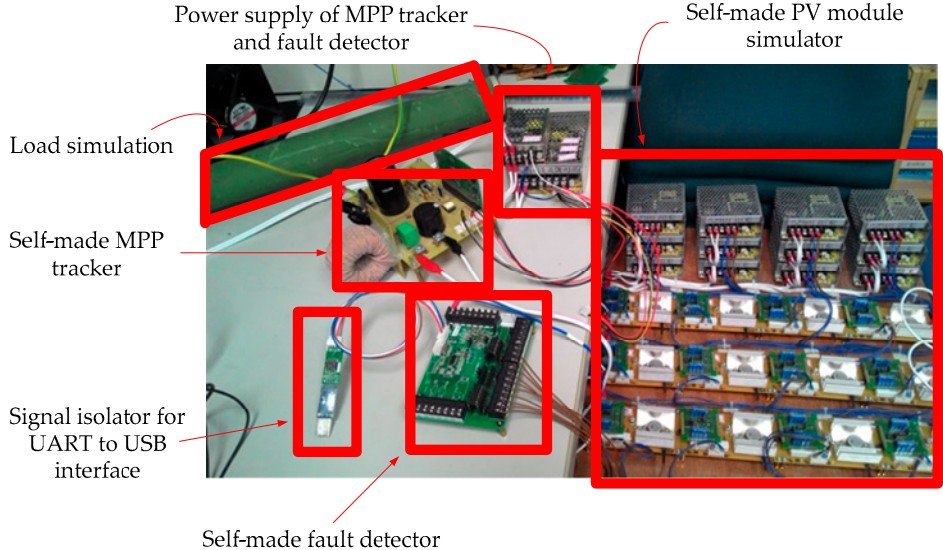

**Figure 15.** Hardware exterior of the proposed fault diagnosis system for PV module.

| Failure type | Failure status |
|---|---|
| PF1 | Normal operation |

| Actual values measured by MP-170 | | |
|---|---|---|
| *Imp*(A) | *Vmp*(V) | *Pmp*(W) |
| 4.52 | 67.76 | 305.95 |
| Actual values measured by PSO-based MPP tracker | | |
| *Imp*(A) | *Vmp*(V) | *Pmp*(W) |
| 4.51 | 66.33 | 300.48 |

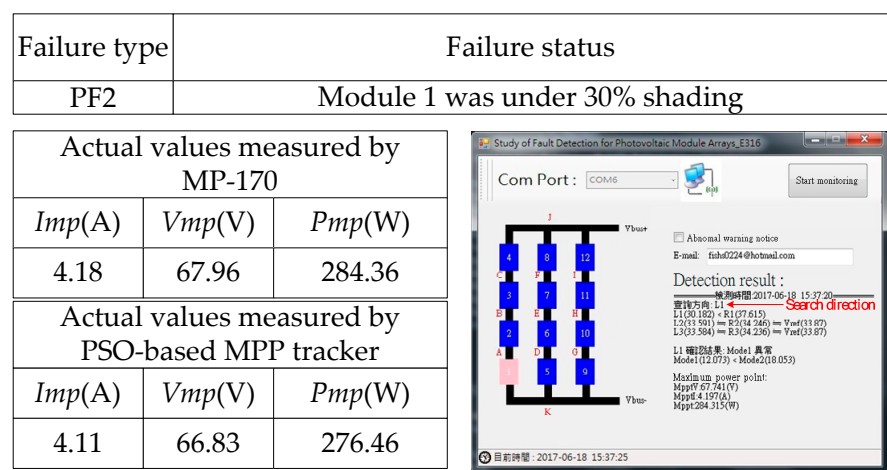

**Figure 16.** Experimental outcome for failure type PF1.

| Failure type | Failure status |
|---|---|
| PF2 | Module 1 was under 30% shading |

| Actual values measured by MP-170 | | |
|---|---|---|
| *Imp*(A) | *Vmp*(V) | *Pmp*(W) |
| 4.18 | 67.96 | 284.36 |
| Actual values measured by PSO-based MPP tracker | | |
| *Imp*(A) | *Vmp*(V) | *Pmp*(W) |
| 4.11 | 66.83 | 276.46 |

**Figure 17.** Experimental outcome for failure type PF2.

| Failure type | Failure status |
|---|---|
| PF3 | Module 7 was under 30% shading |

| Actual values measured by MP-170 | | |
|---|---|---|
| *Imp*(A) | *Vmp*(V) | *Pmp*(W) |
| 4.18 | 68.07 | 284.69 |
| Actual values measured by PSO-based MPP tracker | | |
| *Imp*(A) | *Vmp*(V) | *Pmp*(W) |
| 4.15 | 66.08 | 276.07 |

**Figure 18.** Experimental outcome for failure type PF3.

| Failure type | Failure status |
|---|---|
| PF4 | Module 5 was under 30% shading, whereas module 9 was under 50% of shading |

| Actual values measured by MP-170 | | |
|---|---|---|
| *Imp*(A) | *Vmp*(V) | *Pmp*(W) |
| 3.47 | 69.48 | 240.81 |
| Actual values measured by PSO-based MPP tracker | | |
| *Imp*(A) | *Vmp*(V) | *Pmp*(W) |
| 3.48 | 66.46 | 233.29 |

**Figure 19.** Experimental outcome for failure type PF4.

| Failure type | Failure status |
|---|---|
| PF5 | Module 9 was under 50% shading, whereas module 10 was under 30% shading |

| Actual values measured by MP-170 | | |
|---|---|---|
| *Imp*(A) | *Vmp*(V) | *Pmp*(W) |
| 3.88 | 67.82 | 262.79 |
| Actual values measured by PSO-based MPP tracker | | |
| *Imp*(A) | *Vmp*(V) | *Pmp*(W) |
| 3.82 | 65.96 | 254.39 |

**Figure 20.** Experimental outcome for failure type PF5.

| Failure type | Failure status |
|---|---|
| PF6 | Module 11 was under 30% shading, whereas module 12 was under 50% shading |

| Actual values measured by MP-170 | | |
|---|---|---|
| *Imp*(A) | *Vmp*(V) | *Pmp*(W) |
| 3.88 | 67.72 | 263.02 |
| Actual values measured by PSO-based MPP tracker | | |
| *Imp*(A) | *Vmp*(V) | *Pmp*(W) |
| 3.79 | 66.08 | 253.41 |

**Figure 21.** Experimental outcome for failure type PF6.

| Failure type | Failure status |
|---|---|
| PF7 | Module 8 was under 30% shading, whereas module 12 was under 50% shading |

| Actual values measured by MP-170 | | |
|---|---|---|
| *Imp*(A) | *Vmp*(V) | *Pmp*(W) |
| 3.53 | 67.88 | 239.88 |
| Actual values measured by PSO-based MPP tracker | | |
| *Imp*(A) | *Vmp*(V) | *Pmp*(W) |
| 3.44 | 66.58 | 232.32 |

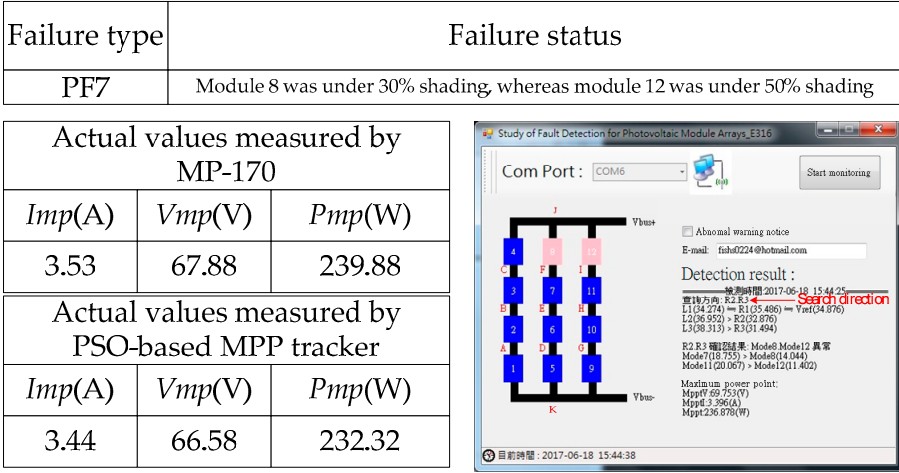

**Figure 22.** Experimental outcome of failure type PF7.

| Failure type | Failure status |
|---|---|
| PF8 | Module 4, 8, and 12 are under 30% shading |

| Actual values measured by MP-170 | | |
|---|---|---|
| *Imp*(A) | *Vmp*(V) | *Pmp*(W) |
| 3.58 | 64.27 | 230.35 |
| Actual values measured by PSO-based MPP tracker | | |
| *Imp*(A) | *Vmp*(V) | *Pmp*(W) |
| 3.34 | 68.22 | 229.78 |

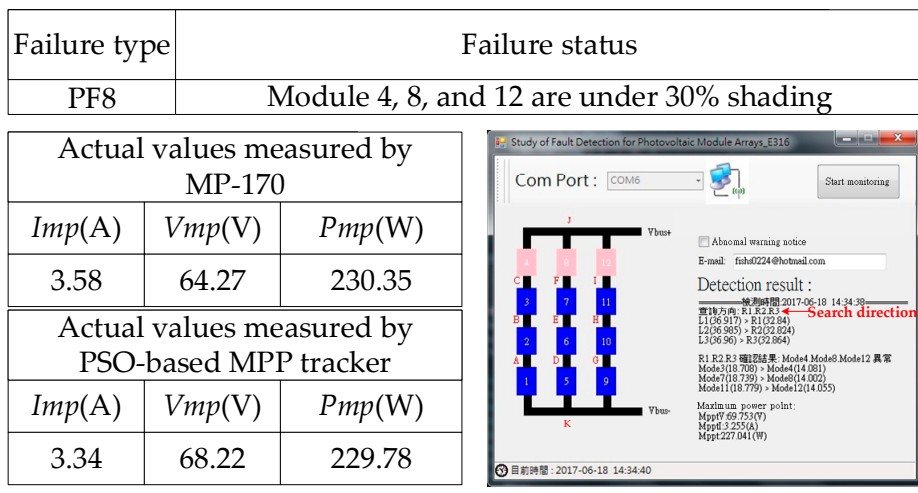

**Figure 23.** Experimental outcome for failure type PF8.

| Failure type | Failure status |
|---|---|
| PF9 | All modules are under 30% shading |

| Actual values measured by MP-170 | | |
|---|---|---|
| *Imp*(A) | *Vmp*(V) | *Pmp*(W) |
| 3.20 | 71.56 | 229.21 |
| Actual values measured by PSO-based MPP tracker | | |
| *Imp*(A) | *Vmp*(V) | *Pmp*(W) |
| 3.15 | 69.10 | 220.40 |

**Figure 24.** Experimental outcome for failure type PF9.

To display the superiority of the proposed binary search-based fault detection method, the extension neural network (ENN) [15], fuzzy neural network (FNN) [21] and multilayer perceptrons and back propagations (MLP) [22] with three network structures, namely 4-7-10, 4-8-10, and 4-9-10 (input layer-hidden layer-output layer) were constructed to diagnose photovoltaic module array faults.

The comparison results shown in Table 2 indicate that a binary search-based fault detection method conducting fault diagnosis is highly accurate and requires minimal iterations, and is therefore, more efficient than the existing neural network methods.

**Table 2.** Comparison of fault diagnosis accuracy rates when applied in a binary search-based method and the existing neural network methods.

| Method / Comparison Item | Iteration Number | Learning Accuracy Rate | Diagnosis Accuracy Rate |
|---|---|---|---|
| Proposed method | 0 | 100% | 100% |
| ENN [15] | 22 | 100% | 99.43% |
| MLP(4-7-10) [21] | 8507 | 90.85% | 93.33% |
| MLP(4-8-10) [21] | 11,089 | 85.65% | 90% |
| MLP(4-9-10) [21] | 8597 | 96.64% | 93.33% |
| FNN [22] | 5584 | 98.84% | 97.70% |

In addition, the proposed binary search-based method only needs basic computing rules such as addition and subtraction. Compared with the ENN presented in Reference [15], MLP method presented in Reference [21] and FNN proposed in Reference [22], the learning and diagnosis response of the proposed binary search-based method is also faster under the same conditions. It is because more complex computing instructions such as exponential function are needed using the existing ENN, MLP and FNN methods. However, the proposed binary search-based fault detection method does not need a complex learning procedure, such that it can easily be implemented on a single-chip microcontroller. When the capacity of photovoltaic modules increases or reduces, no data should be modified to promote the diagnosis accuracy. Because the diagnosis scheme is simple, a PIC microcontroller can be utilized to implement the hardware for real-time fault diagnosis of a photovoltaic module array. There are a lot of analog and digital modules in the PIC microcontroller, so it makes the hardware circuit size very small and reduces cost.

## 6. Conclusions

This study examined various failure types and demonstrated that integrating PSO with the conventional perturb-and-observe method can accurately track the MPP of PV modules. The feature of the fault detector developed in this study is that the sampling channel can be switched by the analog switch for testing PV modules. Therefore, the microcontroller requires only a single set of A/D sampling pins, which can thus reduce hardware circuit construction costs. In addition, the proposed system applies a binary search algorithm to determine the search direction for locating abnormal modules in order to improve the search speed and efficiency. The diagnostic results can be displayed on a PC through a platform implemented using Visual C#. The test information can also be delivered to the designated e-mail address through the Internet. Various failure types developed by the simulator were assessed experimentally, and the results indicate that the proposed fault detector can accurately locate faulty modules.

**Author Contributions:** The conceptualization was proposed by K.-H.C., who also was responsible for writing—review & editing this paper. Y.-H.C. completed the formal analysis of the fault detection based on binary search algorithm. J.-H.T. carried out the data curation, software program and experimental validation. K.-H.C. was in charge of project administration.

**Funding:** This research was funded by the Ministry of Science and Technology, Taiwan, under the Grant Number MOST 106-2221-E-167-013-MY2.

**Acknowledgments:** The authors gratefully acknowledge the support of the Ministry of Science and Technology, Taiwan, under the Grant Number MOST 106-2221-E-167-013-MY2.

**Conflicts of Interest:** The authors of the manuscript declare that there is no conflict of interest with any of the commercial identities mentioned in the manuscript.

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
