# Peer review of "Development of a Low-Cost Fault Detector for Photovoltaic Module Array"

_electronics, doi:10.3390/electronics8020255_

Reviewer 1 Report

Dear Authors,

Thank you for taking the to prepare a manuscript to address the fault detection of PV cells.

My comments are:

The presentation, graphics and English -all are very fluid and clear. 

The use of binary search for fault location is not novel, but application in PV cells' abnormality is interesting.

In Line 173, usage of 8-bit Micro-controller is mentioned. Authors may mention the model for the said device for reference for others.

In Fig 8 and 9, faults at the upper modules (4,8 and 12) are displayed. To show the functionality and capability of fault detection at random, a fault case like in module 1,7 and 10. In Fig 19 to Fig 23, shows faults in either same row or same column.

In Fig 11, at 0% shade,the self-made simulator shows negative delta (underestimation). In Fig 12, at 30% shade, the self-made simulator shows positive delta (overestimation). Can the authors shine some light on this behavior?

Overall, the paper covers the topic with interesting presentations. 

I would recommend this manuscript for publication.

Sincerely,
The Reviewer

Author Response

The authors are grateful to anonymous reviewer whose constructive comments and valuable suggestions have made substantial improvements on this revised manuscript. In order to improve the readability and clarity of this paper, the authors have done their best to appropriately revise it according to theses directions and suggestions. Attached file is a listed response to the reviewer's comments.

Reviewer 2 Report

This paper presents a detailed study on the binary-search-based fault detection for PV module arrays. This paper is clearly written, but the authors are suggested to make the technical contributions more solid by addressing the following two concerns:

1) Highlight the technical contributions by comparing with the existing methods, especially by means of simulation or hardware experiments;

2) Generalize the technical statement of the proposed method to PV module arrays of a larger size (not limited to the fixed size of 4 times 3).

Author Response

(The authors gave the same response as above.)

Reviewer 3 Report

Line 76 to ensuring -> ensuring

Line 89 is shown in Figure

Line 127 as shown in Figure

Line 156 is shown in Figure

Figure 4 improve position of No (close to line)

Line 162 second part of the sentence is hard to understand

Line 194 start new subsection on next page

Line 232 no comma

Line 264 start with the conclusion (section 6) on next page

In the reference list some years of publication appear in bold type, others in normal type; please correct.

Author Response

(The authors gave the same response as above.)
